# New Paradigms in French Historiography, or the Same Old Ones?

## Monica Martinat

Rhône Historical Research Laboratory (LARHRA), Université Lumière Lyon 2, 69007 Lyon, France;
monica.martinat@univ-lyon2.fr

**Abstract:** This article presents some recent trends in French historiography that concern the relationship between history and literature. Among the recent developments are "experiments" carried out by a few historians, which are characterized by an explicit determination to focus on narrative, along with a willingness to share one's own historical subjectivity. By going through some of the examples from this approach, this article highlights how these literary reflexes make important contributions. However, it also points out the weakness of this proposed method of making history on epistemological grounds. That is, it abandons the form of historical writing that requires distance and an appreciation that history's vocation is to propose solid but uncertain propositions (to paraphrase Zemon Davis). By insisting on emotional and sensitive understanding, the knowledge gained from these experiments only questions the scientific aspects of history and history itself. This recent trend is not exactly new, as it evidently links up with some of the consequences generated by the linguistic turn.

**Keywords:** French historiography; historical epistemology; history and literature; linguistic turn





## 1. Introduction

A recent book by Sabina Loriga and Jacques Revel reviews the history of the linguistic turn and its impact on the way history is made in different parts of the world. The authors underline the multiple ways in which this crucial turn has impacted the humanities since the 1970s. Radical in its desire to make history a text without any external referent, or more nuanced, this movement has imposed a way of formulating the questions of reality, truth, and the historiographical operation on different generations of historians, also marked by a specific apprehension of the more general questions of their time. The book presents—in a fine and intelligent way—the varied and sometimes contradictory character of the linguistic turn; it also restores historical depth by linking it not only to its claimed ancestors, but to historiographic reflections that have been present since the 18th century. It concludes by denying that the linguistic turn had any influence on historiography in Europe, and particularly not in France:

"In Western Europe, with the exception of Great Britain where it has provoked . . . vigorous debate, the linguistic turn has hardly been considered. It has been, at best, the object of critical suspicion in Italy and Germany, for example, and the object of a resolute refusal by the great majority of French historians. These refusals are the result of a diversity of reasons: the installed power of social history, which felt its approaches and ambitions challenged; the existence of contemporary proposals that undertook, on their own account, a critical reflection on the certainties and practices of the profession; the lasting imprint of a positivist tradition quick to denounce the proclamations of the linguistic turn and their further possible outcomes; the reticence in France . . . to see the discipline lurch towards theoretical formulations that could evoke philosophies of history. And also, no doubt, an anti-Americanism that is rarely made explicit, at a time when

the center of gravity of historiographic production and innovation seems to have moved to the other side of the Atlantic" (Loriga and Revel 2022). The authors of the book are right when they note the relative absence of a theoretical debate on the philosophy (or philosophies) underlying the linguistic turn, and they are also right in underlining how difficult it is to give a unequivocal sense to this turn and to the meaning that it had within the historian's milieu. As Christian Delacroix pointed out a decade ago, historians focused on two different aspects: the role of language in building identities and social realities and, overall, the skeptical and relativist implications of its propositions, overlapping in this respect with postmodernism (Delacroix 2010; White 2010). Rather than reinforcing scientific rigor, we can say that in history it had the opposite effect: this is one of the first and most radical criticisms made of it by Arnaldo Momigliano and Carlo Ginzburg, as far back as the 1980s[1].

It is not my intention here to open a full discussion on the meaning of the linguistic turn for French historians, or to clearly distinguish its reception in historiography from its original intention or the meaning it had in (and for) other fields. I only intend to suggest that, in some current historiographical trends, we can see a delayed and devious inheritance of the potential weakening of historical truth—or the indifference towards it—that comes from the historiographical interpretation of the linguistic turn. Contrary to their claims, they do not introduce a new paradigm, but take up relatively old positions, moving the discussion from the truthfulness of history to the more discipline-neutral ground of its relationship with literature. Some current historiographical trends that point out the similarity between historiography and literary work, and that aim to build strong bridges between the work of historians and (mainly) novelists, are indeed influenced by a weak conception of truth and by a certain indifference towards the reality underlying historical texts. They also have a specific attitude towards documents—a kind of fetishistic one; they also tend to place historians at center stage, along with their subjectivity and even their emotions. Thus, perhaps in spite of the historians who represent these trends themselves, they end up preventing the historiographic operation from being the bearer of a potentially shared interpretation of the past, the universality of which would be linked to the specific nature of history as critical knowledge. They suggest a specific paradigm for history that is rooted in revisiting certain elements derived from the *linguistic turn*.

These are historiographical currents that are neither structured nor in the majority (based on a look at a summary of academic historiographical production); however, they have received—and are receiving—quite a bit of editorial and media attention, putting them in a position of visibility that deserves to be questioned. My purpose is to present these trends and show that we are not dealing here with a new paradigm, as many proponents of these currents suggest, but with the permanence and revisitation of an old one that has greatly contributed to weakening the historical enterprise since the 1970s.

## 2. "Littératures du Reel" (Jablonka 2014)

At the end of the first decade of this century, a literary period dedicated to history emerged in France. Anticipated in 2006 by Jonathan Littell's novel, *Les Bienveillantes* a wave of novels dedicated to more or less recent history swept throughout the country. The media insisted on talking about a *literature* under the imprimatur of "history." The topics covered included the Algerian War, the Shoah, the Second World War, and the Great War. These great events have been taken over by literary fiction that has shown itself to be very effective in producing historical knowledge of facts and, moreover, in providing depth and meaning to the past[2]. This recourse to narrative—and to a "competent" narrative—is not only a challenge to the traditional production of historians; it is also supported by the Aristotelian claim that views literary fiction as superior to history because of its universality. The meaning of the past has been measured by the capacity of a narrative to resurrect, and to arouse, emotions among readers—especially emotions of a potentially universal nature.

This literary fiction has carved out a specific space for itself, filling that no man's land that lies between memory and history: the protagonists of these recent parts of history were disappearing while not all of the archives were yet accessible to historians. "Who bears witness for the witness?"—a reworked quotation from Paul Celan that Yannick Haenel puts in the title of his novel (Haenel 2009)—is a real program from a good number of writers of this period. Historians rightly see it as a challenge on their own grounds, and they smell defeat.

Faced with the loss of hegemony over the narratives of the past, historians have reacted in two ways: one is to publicly question and confront the value of literature and its role in the transmission of history[3]; the other is to experiment. These experiments are part of a reflection that history has always had on its own nature and on the formal ways best suited to present the contents of its work. However, at this particular conjuncture, these ways are also a part of a successful resurgence of a fictional literature that claims to have a historical vocation, and deliberately proposes to do history and to tell the truth about the past—truth that historians would be unable to tell because of the intrinsically political nature of the discipline[4].

Among literary novelists one finds both the will to make the past the substratum of an invention capable of rewriting history in a more truthful way (Haenel), and that of restoring the facts intact and emptied of all invention (Binet); among historians, on the other hand, one finds the temptation to emancipate oneself from the rules of historical writing. The upshot of this approach is that we can find a few formal changes in the presentation of historical works, such as the lightening of or even the elimination of the critical tools and, in particular, footnotes (which are progressively removed from books to improve their readability), but also more substantial ones, such as the abandonment of scientific objectivity in favor of a subjectivity that makes the authority of the historian the main criterion of validation. These are just two examples in which one sees an interesting transformation of contemporary historiography that aims to situate itself on a terrain close to, if not identical with, that of literary creation.

Ivan Jablonka, a historian who writes under his own name and a fiction writer who used to write under a pseudonym, probably best embodies this tendency, which he makes explicit in one of his latest works after having theorized about the integration of history into literature in an essay intended as a political proposal. That is, in *L'Histoire est une littérature contemporaine: Manifeste pour les sciences sociales*, Jablonka seeks to re-establish the dialogue between the social sciences and a wider audience, through a recomposition of different genres in a new category—that of the "literature of reality" (*littérature du réel*), which includes history, literature, memoirs, testimonies, etc. (Jablonka 2014). In his view, the requirements of a piece of writing stripped of its academic appearance are put at the center of this creation, which advocates for a readable genre, where style is put at the service not only of the narrative but also of the (historical) discourse. According to Jablonka, this is one tool among many to better achieve the goals that social scientists set for themselves. It also represents a new freedom for historians.

The features of this new freedom, however, raise questions about the means of scientific validation of the historical discourse. The subjectivity of the writer, i.e. his or her life experience, is paradoxically the basis for asking the reader to trust the writer's historical interpretation. The academic title of the historian serves then as a real guarantee, without need for him to prove that the set-up on which he delivers himself, corresponds to some requirement of external truth. The reader ends up suspending incredulity in a way appropriate to literary fiction. Just because literature claims to be based on historical research and facts does not change the contract between writer and reader, which rests on the possibility of fiction. The literary historians have put themselves in a similar position. A "confusion of genres" is certainly not new. One need only think of the emergence of the historical novel on the one hand and the contemporary historical production of a Michelet on the other, of the "factographies" of the 1920s (Zenetti 2014), or of Truman Capote's non-fiction novel (Capote 1965), of the "return to narrative" of the historians of the 1970s,

and so on (Stone 1979). So, this recent new wave of "investigative literature" (Demanze 2019) reopens a very old file—that of the way in which literature, including fiction, allows us to grasp reality. It is a response to the criticisms brought to the capacity of the literature to tell some kind of historical truth. History tries to respond to other criticisms: at best to be frozen in a supposedly objective and impersonal narrative, without any real hold on the readers, and at worst to not know how to restore the past in an efficient way, in good part because it maintains an academic language that overplays objectivation. One can see that attempts at literarizing history amount to a proposal, from various current researchers, for restoring history with its explanatory qualities, along with its power to be the bearer of a truthful discourse on the past. In short, historians surf on the recognized capacity of literature as a form of knowledge by insisting on the power and the performativity of the (literary) language that they then appropriate. At the heart of this assimilation is the claim of authoritativeness from the historian who gives himself the right to interpret the past through an all-powerful "I". But what is the meaning of this new "I" that we find at the center of these historical experiences? If it is a resource for taking scientific history out of its academic framework, which is now viewed as old and almost anachronistic, it also ends up signifying the impossibility of proposing shared and collective interpretations of the past [5] and of trivializing its systems of proof. What becomes important is to penetrate the reader's emotions, to create a sensation, to produce empathy with the past (near or distant), and to make it "come alive", even if it means juxtaposing very diverse objects that systematically confuse the reader as to their status as truth.

Consider an artistic operation, such as the one undertaken by the online journal *Entre-temps*—a "digital journal of current history, collective and free, attached to the chair of *History of Powers in Western Europe, 13th–16th century* of Patrick Boucheron"[6]. It contains a strange mixture of articles, including historical or sociological analyses, interviews, and also sections that play the game of possibilities. For example, the magazine launches a game, which it describes in a synthetic way as follows: "The magazine *Entre-Temps* launches today a new game of collective writing of history. The starting point is a photograph of a bare parcel of land (or building) in a large European city. The aim of the game: to write the history of the habitat and of the inhabitants who might have occupied it since Antiquity". Historians—mostly young ones—take the journal up on its challenge by writing short, imaginative articles, accompanied by a photo of the trace that they have chosen. This game is part of the journal section called "*Transmit*", which is aware that the pedagogy of current history renews the very meaning of history[7]. We can now imagine that both invention and the paraliterary experience are part of the pedagogy of history, ultimately putting the search for truth and the expression of the fictional on the same plane[8].

It should be pointed out that this movement of collective imagination is not new, even if it has taken a new form in recent years. A book published in 2008 and signed by Philippe Artières, Anne-Emmanuelle De Martini, Dominique Kalifa, Stéphane Michonneau, and Sylvaine Venayre proposed, in playful terms, to confront the readings and uses that five historians, specialized in different fields, had made of a genuine file containing information about a Mr. Bertrand bought by chance by one of them at a flea market (Artières et al. 2008). The book ends with a "Table of Discordances", which lists the various readings proposed by the participants that are linked to the file's elements—which are, after all, marginal. This includes, among many other things, the title of the folder containing the documents, the extreme dates found in the documents, the date of birth of Mr. Bertrand, and his first name. The documentary deconstruction is such that it casts doubt on the ability of historians to establish a common version based on documentary verification and, if necessary, on subsequent research that makes it possible to distinguish truth from error; it is an end in itself, exclusively serving the demonstration of the game. The "document" is placed at the center of an investigation that cares little about going beyond it.

### 3. Fetishism of Archives[9]

We thus arrive at a paradox: on the one hand, the notion of the document is criticized and called into question because it is the bearer of errors and power relations, the recognition of which is one of the main goals of historians; on the other hand, it is used as an inescapable marker of the true existence of the object it reflects, i.e., in the case cited above, of a Mr. Bertrand, whose real existence is necessary for the game to take place. We should perhaps also ask ourselves whether the presence of photographs reproducing documents and accompanying historical writing—especially when the narrative deviates from the beaten path of impersonality by taking on the clothes of an assertive subjectivity—aims to claim authenticity. It can leave the impression that the authors need to reinforce the proof that the narrative draws its *raison d'être* from reality, that it is anchored in it, and that it is not only the result of the author's imagination. Indeed, the archival quotation, even in a photographic way, provides an effect that engenders reality (Martinat 2020). In fact, it is also a process used deliberately by novelists to confuse the reader and the relationship between their text and reality. A recent example, perhaps clearer than others, is Camille de Toledo's beautiful book *Thésée, sa vie nouvelle*—an autobiographical account that is not designed to be faithful to factual reality; rather, it offers photos and documents with ambiguous status, serving as proof of reality, but also blurring the lines between fiction and reality[10]. Here, we are clearly in the field of literary fiction, where this blurring is consubstantial and can also prove necessary. The use of this mechanism by historians today seems to be in large part a literary borrowing. It is in this context that I view the example proposed above, concerning the game organized by the magazine *Entre-temps*. It is in fact part of a "playful" approach to history that makes the archives the primary site of historical experimentation, but it does not seem to be very interested in saying something true about reality.

The publisher *La Découverte* recently opened up a collection entitled "*À la source*" as an exemplary way to present a new type of relationship between historians and archives:

"*À la source*: Where those who try to understand the past and strive to weave the story drawn from it. There, where the pleasure but also sometimes the embarrassment of the discovery is felt. Rare, unusual or dissonant, certain pieces of archive material intrigue us, disturb us, leave us unsettled. What to do with these heady sources? Don't they often open up unexplored paths? *À la source* invites historians to take a close look at them. Objects, photographs, engravings, prints, manuscripts: these sources—reproduced or retranscribed—are here the center of gravity for a writing experience that, while following the historical method, frees itself from conventional narrative frameworks and assumes a sensitive approach to the past"[11].

Once again, a game guides the writing: the one that invites the historian, starting from a document that disturbs and that becomes the "pretext" to carry out an investigation that must also be accompanied by a particular writing, free from the academic straitjacket, to say the least. The historians who have published in this collection so far have done so in very different ways, each following his own thread. The source is nevertheless central, as it puts itself at the heart of the historian's work: the source, not (necessarily) the reality to which it refers. It is the relationship that the historian weaves with the source that is critical, not the complex and global result of the work of constructing a historical interpretation. Locating and consulting archives becomes part of a narrative whose focus is turned on the papers that attest, rather than on what is attested; its focus is on the text, rather than on the reality of which it is only the trace. Zadig would be lost in detailing the shape of the trace left by the queen's dog, rather than in deducing from it the passage of an animal in flesh and blood[12]. The trace is the only possible reality to which the historian can have access, and all his or her efforts must therefore be concentrated on his/her ability to dismantle and reconstruct it, as if it were the reality of which he speaks.

We can therefore speak of the archives by underlining the ways in which they affect the historian in an almost sentimental sense of the word. A "sensitive" history, where

the feeling is above all the historian's relationship with the sources, and which is then empathetically transmitted to the readers. Reality then emerges through this emotional register, and we are in a relationship of pre-critical adhesion to the interpretation. The historian does not hesitate to explain the subterranean paths that connects him emotionally or casually to the meaning he gives to his sources, as Jérémie Foa does when he telescopes between a mention that he finds in a document concerning Saint Bartholomew's Day and an event in French current affairs that marks him as a citizen and historian[13]. Arlette Farge, whose work inaugurated the *À la source* collection in 2019, starts from the "waste", or more elegantly from the "relics" of 18th century archives, by proposing these fragments with a minimalist contextualization. As she claims again in the conclusion, they do not teach us anything that we do not already know. However, they move us and allow us to reweave a human and sensitive link with the men and women of the past:

> "To look at these different and distant societies ( . . . ) it is necessary to appeal to what is common to us, women and men of the twenty-first century, and to them. And to mobilize some things that are of the order of the universal allowing understanding. Let us dare to consider in their right place, affect and sensibility as human universals, allowing to be closer to the harshness of the reality of the past centuries and to better seize in their most unexpected moments, these forgotten worlds; here are the shadows of the century of the Enlightenment". ([Farge 2019])

It is unquestionably a form of knowledge that is summoned here, and which has also structured the work program of this historian, very much linked to the exploration of ordinary and popular worlds, since her work with Michel Foucault ([Farge and Foucault 2014]) or her study on the Parisian people published in 1986 and republished in 2016 ([Farge 2016]). The taste for the archive (*le gout de l'archive*) and the emotion that it arouses are the two foundational elements of what she provides here to the reader, with the main task of the historian being, in this context, to make us feel the same emotions that she feels when in contact with her material traces that come to us from the past, through the intermediary of a writing impregnated with literary narrative[14].

The discovery of a "thick bundle of small cards" in a large retirement home that is closing in 2014 is the pretext that pushes Mathilde Rossigneux-Méheust to investigate men and women who, between 1956 and 1980, had been classified as "never to be taken back" by the institution ([Rossigneux-Méheust 2022]). Her narration takes us behind the scenes of the production of stigma that institutions generate by classifying individuals into a "specific paper disciplinary construct"[15]. The historian makes an almost literal interpretation of the constitutive instruction of the collection in which the volume appears, systematically returning to the chosen source after having logically expanded her documentary field, allowing us to see a living and diverse reality. However, these frequent returns to her "bundle of cards" reify the document, giving it a very particular status that makes it almost more important than the reality that she was able to show us, and without managing to pierce the specific reasons that presided over the production of this type of document.

This is perhaps the only work in the collection that accepts the challenge of the exercise. The other volumes seem, on the contrary, to emancipate themselves rather quickly from at least part of the instructions suggested by the director of the collection, Clémentine Vidal-Naquet, i.e., those relating to the source that disturbs and opens up unexplored paths[16]. It is the narrative construction and the sensitive approach to the past that prevails. One of the authors, Hervé Mazurel, is moreover one of the designers and founders of the journal *Sensibilités*, inaugurated in 2016[17]. In this collection, he wrote a book, published in 2020 and dedicated to Kaspar Hauser—one of the most famous "wild children" of the 19th century ([Mazurel 2020]). The subtitle of the book, *Essay of abyssal history and sensitive anthropology*, reveals a program that goes beyond the case studied and responds to a structure that is meant to be like music, with a prelude and a coda instead of the more traditional introduction and conclusion. Moreover, the author introduces himself on the back cover as a historian and musician. The subjectivity of his work is more than assumed

in the inscription of the book in the category of *essais*[18]. The questions that it poses are of a general nature and aim to reflect on the individual and collective significance of an unsocialized being suddenly confronted with a society and a culture that he is unaware of, in content and in form. It also asks an even broader question, this time dedicated to the influence of history on all of us: "Now there could well reside in our eyes all the value of the Hauser case: that of revealing to us, by its very aberration, by contrast, therefore, to what obscure and secret depths the influence of history descends in ourselves"[19].

These few movements in current French historiography do not represent the majority—far from it. Academic production is, for the most part, faithful to traditional canons, both in form and in theme. However, these exceptions are interesting to look at, because their proposals start from a necessity that most historians feel: to renew a stagnant historiography, which takes refuge in an often immobile traditionalism. History is losing momentum; no longer does it have the large place that it once had in the French intellectual panorama, when the international success of the *École des Annales* reigned.

Let us take a look at the stages of a French historian's academic career, marked by formal necessities and by particular procedures: a doctoral dissertation written most of the time with the use of a "we" for objectification, which must successively be transformed into a book, followed later by a *Habilitation à diriger des recherches* that is still under the direction of a supervisor, and which consists of three parts: an original and unpublished work, the whole of a historian's publications and scientific activity, and a text that historians informally call the "ego-history", in which the candidate reveals career and biography in variable ways and in whatever form pleases the author.

One can see how a historian's career path, which includes formal and academic constructions along with a creative freedom expressed through the "ego-history memoir", puts him at the center of an academic scene that serves to better interpret his scientific production. It is this creative freedom component that has arguably inspired the historiographic tendencies that we now see in their experiments. The *Éditions de la Sorbonne*—the editorial structure behind the University Paris 1 Panthéon Sorbonne—started a collection called "*Itinéraries*", which proposes the texts of these essays of "ego-history" that go on to become scientific proposals. The collection was first directed by Patrick Boucheron—a professor in this university before his election to the Collège de France[20]. The Panthéon-Sorbonne University is perhaps the center of this polyform movement that I have tried to present briefly in the preceding pages. Several of the authors quoted here have followed part of their studies in this institution, and all belong to the generation born between 1966 (Anne-Emmanuelle Demartini) and 1977 (Hervé Mazurel). The intellectual center of this renewal of history towards sensitivity and towards experimentation is Alain Corbin, and several of the historians quoted above studied with him. He is the author of an immense bibliography, which has opened up new fields in French historiography—most notably in the history of sensibilities[21]. Between the end of the 20th century and the beginning of the new one, Corbin proposed two works that, in my opinion, constituted a real turning point in the way of making history. In 1998, Flammarion published the book *Le Monde retrouvé de Louis-François Pinagot: sur les traces d'un inconnu (1798–1876),* which was intended as an archival experiment; for Corbin, it was a matter of choosing an individual from the archival documentation about whom very little was known—a "( . . . ) Jean Valjean who would never have stolen bread" (Corbin 1998)—in order to approach as closely as possible the normality *and not the exceptionality* of the lives that the archives often give us to see[22]. The reading between the lines of a limited documentation here is based on forms of generalization that apply to this individual what is known about others. It reveals itself more as a rhetorical expedient than an innovative proposal. One wonders whether this game of self-limitation of the historian in front of its sources, beyond the evocation of an obvious conceptual problem—that of generalization from a case—does not encroach on a terrain that is traditionally and more legitimately literary, where imagination plays a predominant role. The work published in 2011 takes this step and allows the imagination to overflow to the point of deliberate invention. In *Les Conférences de Morterolles*, Corbin imagines (by

inventing them) the texts of public lectures held by a schoolteacher in a small village. In fact, only the themes of the lectures and the number of participants are preserved in the archives—not the texts, which may never have existed (assuming that the speaker spoke without a written text) (Corbin 2011).

The inspiration for the works that I have examined in this article comes from the literary field. They are all flirting with fiction, and they do not even feel the need to differentiate themselves from fiction—they do not claim the specific relationship to the truth that has been seen as proper to history since Aristotle. That history must be dusted off is a fact. That this operation, among other things, passes through writing also seems necessary, not only because it is a way to communicate more clearly to a wider audience, but because writing produces interpretation and meaning. To use the "I", the "we", or the impersonal form determines specific postures. This is not an insignificant gesture, even if one can discuss its reasons and effects (Traverso 2020). The literary posture of historians, however, goes well beyond the narrative resources that they choose to use. As Jablonka explains quite clearly in the title of his book, it is a question of rejecting the borders between literature (including fiction) and history in the name of the common task of producing knowledge out of reality that both propose. However, the knowledge of the world produced by literature and that produced by history are not identical—the two mobilize different resources. It seems to me that the destruction of the frontier between history and literature by historians engenders a loss, not a gain. We lose appreciation of a critical approach that requires an *in*sensitive relationship to the sources, and we no longer compel the reader to draw an interpretation that is more intellectual than empathetic.

Literature takes us, as readers, to a different terrain—one where the border between what is true and what is fictional no longer necessarily matters, and where our adherence to the narrative is a sensitive, empathetic adherence, based also on the specific pact that binds us to the author. The historian's pact with his readers is to tell the truth and to bring them to a critical understanding of the construction of the interpretation that necessarily lies behind the historical operation. Historical interpretations are fragile and should be presented in a hypothetical way. Natalie Zemon Davis has said it with intelligence: the strength of history is a weak strength, which consists in proposing solid but uncertain interpretations, made of "perhaps" and evoked hypotheses that are not necessarily confirmed, but they could be explored, checked and, ultimately, possibly validated[23]. Literature confronts us with an ultimatum—one either enters the author's game or one does not; one is unable to disassemble the author's interpretation from the adequacy of the author's documented reality. History can suggest a more open form of knowledge. A problem then arises not only when historians set aside the requirement to prove and explain their path of proof, but also when they put it under the sign of their sensitivity or their experience. Being a historian is not enough to ensure interpretations[24]. This is where the relationship with the archives, which is at the heart of our work, becomes crucial. That is, we have methods and expertise concerning the handling of traces from the past that need to be put at the service of the search for a truth, and not only to discuss the nature and the conditions of production of the trace itself. While certainly interpretative, what is shared emerges not through sentimental or literary evocation, but through the construction of a shared consensus and a permanent criticism of the sources and effects of the narrative.

Enzo Traverso, in particular, has well captured the "epistemological displacement" that these postures entail. He criticizes "the permanent ostentation of historical subjectivity in a narrative that mixes present and past, and in which the narrator occupies the same place as the actors of history"[25]. He surmises that this tendency is a consequence and a symptom of the individualistic habits produced by neoliberal society. In this respect, historians are put on the same level as novelists who tackle subjective historical reconstruction, such as Daniel Mendelsohn or Javier Cercas—or their original model, W.G. Sebald.

Perhaps subjectivist readings of the world are to be attributed to such a neoliberal turn; however, I am not convinced that the attitudes of novelists and historians should be considered equivalent. The former group strives to measure itself with history, with the

constraint of the documents and their zones of invisibility, while reworking them in a free manner. Thus, they contribute to a specific literary form of knowledge that is necessary for understanding the world better. In contrast, historians, by opting for this literary and subjective posture, emancipate themselves from the constraints of their profession and thereby transgress the founding pact of their social function. Thus, they abandon the eminently critical and distant work that is the hallmark of historians. If the proximity in the postures of the two adapts well to a reception based on the same individualistic presuppositions that make them particularly appreciated, it erases the epistemological difference between literature and history, without benefiting either. In such a melding, knowledge will only be sentimental, emotional, and empathetic. The inconvenient truth is that, according to such a deflationary view, the relationship between the historian and the past—made through the mediation of sources, and with a view to creating reasoned and collective consent—becomes doubly inaccessible and is no longer part of the horizon of our work.

Clearly, the *linguistic turn* has come and left its mark on French historiography, at least in weakening the importance of the relationship between the document and the reality behind it, in insisting on the rhetorical side of historiography, and in pointing out the similarities between literature and history in epistemic terms. In these recent historical trends, we face a new combination of pre-existing elements, such as the importance of the narrative side of historical work, the focus on the archive itself, and the probabilistic nature of historical interpretations, along with some other elements—e.g., the centrality that it places on the historian and his/her feelings, on experimentation, and on a playful side—that are newsworthy. This combination does not provide a new scientific paradigm but, in my opinion, reinforces an old one, proposed a few decades ago by the linguistic turn.

**Funding:** This research received no external funding.

**Institutional Review Board Statement:** Not applicable.

**Informed Consent Statement:** Not applicable.

**Data Availability Statement:** Not applicable.

**Acknowledgments:** The author wish to thank the anonymous reviewers for their work and the relevant comments to the first draft of this article.

**Conflicts of Interest:** The author declares no conflict of interest.

## Notes

1     For a full development on this point and a rich bibliography, cf. Loriga and Revel, *Une histoire inquiète*.

2     Among the books published in 2009, we can mention (Binet 2009; Haenel 2009; Mauvignier2009). In 2013, dedicated to the First World War: (Lemaitre 2013).

3     Two journals have devoted monographic issues to the dialogue between historians and novelists on the relationship between history and literature: *Les Annales HSS* (n°2, 2010) and *Le Débat* (n°165, 2011).

4     This is the position assumed especially by Haenel in his novel referred to earlier.

5     The practice of using "I" is not new either. Even in French historiography, which is more formal than other historiographies, it has been used for some time. However, it should be noted that it was not designed to point to the historian's omnipotence and authority, but rather to the fragility of his or her interpretation. Jablonka's suggestion is of another kind: it results in putting the historian in an authoritarian position (Traverso 2020).

6     https://www.college-de-france.fr/entre-temps, consulted on 19 December 2022. Here is the detailed presentation of the journal: "*Entre-Temps* is a digital journal of current history, collective and entirely free, attached to the chair of Patrick Boucheron, at the *Collège de France*, inaugurated in October 2018. *Entre-Temps* is a public service of history taking the form of an open space, dedicated to a plural, joyful, interdisciplinary and intermediary history. It is a space for exchange, debate, creation and production. *Entre-Temps* aims to put forward diverse contents (exclusive or collected on the internet). We seek to make visible the diversity and inventiveness of new forms of writing history by identifying them, promoting them and connecting them together. One of the challenges of Entre-Temps is to offer a rich and diverse look at the ways in which history is constructed and deployed. Our journal offers a look behind the scenes, at the "how it's done" of research, writing and its dissemination. It focuses less on "finished products" than on the paths taken, the approaches followed, the methods unfolded. This is one of its original features, which makes it a public service of history: it allows everyone to discover how an issue and an object are made, whether they

are books, films, exhibitions or works of art. The dynamics and the process are as essential as the results. Entre-Temps is an en-cours of history, a journey in its action in the present. In the way of showing the building sites and the making of the buildings, the intermediary dimension is crucial. A singular axis of *Entre-Temps* is indeed to give birth to and restore dialogues between different universes, but which all take history as their object. They may be researchers, teachers, archivists, writers, painters or visual artists, filmmakers and documentary filmmakers. What is essential is their common work, their exchanges and their complementarities, sometimes even, why not, in the tensions that can arise from the divergence of their approaches and their centers of interest".

[7] This is how the section is presented: "Pedagogies of history. Thinking about the teaching of history today in France and abroad. The actuality of pedagogical approaches in history, in all fields—secondary and higher education, cultural and museum institutions, heritage actions, popularization—does not cease to redefine the historical discipline, its reception and the means of transmitting it. ( . . . )", https://entre-temps.net/les-hist-orateurs-nouveaux-transmetteurs-de-lhistoire-sur-youtube/, consulted on 20 December 2022.

[8] This is also what happens in children's literature, which uses fictional characters from various periods of history to introduce people to these periods. On this subject, cf. (Ferrier 2013). Special issue dedicated to "Le récit entre fiction et réalité. Confusion de genres ", n° 2, 2013, pp. 51–61.

[9] This expression came up in a conversation with Mathieu Grenet who, if I remember correctly, was the first of us to use it. I took the liberty of borrowing it from him.

[10] On this book and its autobiographical aspects, see the article by Claire Paulian, "La Fabrique d'une légende," En attendant Nadeau, 18 November 2020, https://www.en-attendant-nadeau.fr/2020/11/18/fabrique-legende-toledo/ accessed on 20 December 2022 (de Toledo 2020).

[11] Presentation text included in all volumes of the collection—six to date (December 2022).

[12] The reference here is to Voltaire's *Zadig*, which Carlo Ginzburg used as one example of his "*paradigme indiciaire*" (cf. Ginzburg 1980). In Ginzburg's view, the *historical trace* is consubstantial with the existence and importance of the reality beyond. This is not the case for the experiments/games that we are discussing here—not because they deny reality, but because the trace is treated as less important than the game itself.

[13] Here is the passage, p. 12: "Among the hundreds of lives lost in the *Mémoires de l'Estat de France*, this singular story is traumatic in its brevity: *Commissioner Aubert, living in the rue Simon le Franc near the Maubué fountain, thanked the murderers who had slaughtered his wife.* A man who thanks the murderers of his wife weighs heavily, and in the flood of baseness that night, this thank you does not pass. Perhaps it echoes too much for me the memory of Françoise, executed at her home one cold morning in November 1991 in Clermont-Ferrand. The husband had paid the killers. The historical montage, I agree, is unexpected—the year 1572 in Paris and the Auvergne of my childhood, the Saint Bartholomew's Day and the 'Ndrangheta. But after all, the past makes a sign as it can and clings where it wants" (Foa 2021).

[14] This "taste for the archive" was the subject of a short book, published in 1989 in the collection "La librairie du XXIe siècle", directed at Le Seuil by Maurice Olender—a particularly important figure in the cultural landscape since the foundation of the review *Le Genre Humain* (1981) and *La Librairie du XXe siècle* (1989), which later became *La Librairie du XXIe siècle.* There is undoubtedly much to be learned about this personality and the fundamental mark that he has made on historical culture (and, more generally, on the social sciences) through his publishing activity. Cf. (Farge 2021). Here is a very significant passage for our purposes (p. 16–17): "A cloth under the fingers: rough softness unusual for hands accustomed by now to the cold of winter. White and solid linen, slipped between two sheets, covered with a beautiful and firm writing: it is a letter. One understands that it is about a prisoner of the Bastille, since a long time locked up. He writes to his wife an imploring and affectionate missive. He takes advantage of sending his clothes to the laundry to insert this message. Anxious about the result, he asks his laundress to embroider a tiny blue cross on one of his cleaned stockings; this will be for him the reassuring sign that his fabric missive Found in the archives, the piece of cloth alone says that there was certainly no little blue cross embroidered on the prisoner's laundered stocking . . . ".

[15] Rossigneux-Méheust. Back cover.

[16] Of the five books published to date, I have not closely explored the books by Hélène Dumas (*Sans ciel ni terre. Paroles orphelines du génocide des Tutsis (1994–2006)*, published in 2020) and Jacqueline Carroy and Marc Renneville (Mourir d'amour. Autopsie d'un imaginaire criminel, published in 2022).

[17] Also part of this quartet is Clémentine Vidal-Naquet, the others being Quentin Deluermoz and Christophe Granger. The latter, however, is no longer a member of the editorial board—which, in addition to the three mentioned, includes Thomas W. Dodman and Anouche Kunt. https://www-cairn-info.inshs.bib.cnrs.fr/revue-sensibilites.htm?contenu=apropos (accessed on 5 January 2023).

[18] What is less clear, in my eyes, is the definition of abyssal history that appears in the title.

[19] «Or là pourrait bien résider à nos yeux toute la valeur du cas Hauser: celui de nous révéler, par son aberration même, par contraste, donc, jusqu'à quelles obscures et secrètes profondeurs descend en nous-mêmes l'influence de l'histoire», (Mazurel 2020), *Kaspar l'obscur ou L'enfant de la nuit (✝ 1833)*, p. 271.

[20] It was then taken over by Yann Potin.

21    It can be found in the *Wikipedia* page dedicated to him: https://fr.wikipedia.org/wiki/Alain_Corbin, consulted on 28 December 2022. Corbin co-signed in 2022, with Hervé Mazurel, the direction of a *Histoire des sensibilités* published in the collection *La vie des idées* at the *Presses Universitaires de France* (PUF).

22    It is a very particular reinterpretation of micro-history "à l'italienne", which had proposed the notion of "exceptional-normal" (Edoardo Grendi) in reference to exceptional cases documented by archives that capture ordinary people at the moment they are in the grip of institutions (often judicial) and, thus, deviate from "normality" while allowing us to glimpse it with controlled forms of generalization.

23    These are considerations that the historian makes in the preface of her book dedicated to the case of Martin Guerre, which she had studied after having been a consultant for the film by Jean-Claude Carrière. cf. (Zemon Davis 1997).

24    Martinat, "Historiens et littérature".

25    Traverso, *Passés singuliers*, 113. Speaking of Jablonka, Traverso says: "( . . . ) the methodological innovation of this writing of history is far from being exclusively stylistic or aesthetic. Basically, what it reveals is an epistemological shift. If historians have always explored and interpreted the past with the more or less sophisticated tools of their discipline, now they do it starting from a subjective interrogation. Now, their books do not only try to answer the question: what happened, how and why, but also—or rather—to another question of an existential nature: who am I, where do I come from, what family or generational links connect me to the past? These considerations apply, in my opinion, to all these "subjective" experiences, even where the links that connect us to the past are not familial or generational".

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
