# Peer review of "New Paradigms in French Historiography, or the Same Old Ones?"

_2410-9789, doi:10.3390/literature3020016_

Round 1

Reviewer 1 Report

You make excellent points about the subject of whether there are new paradigms in French historiography, with apt comparisons to the literary field. In your references to "littératures du réel," I wonder why you didn't reach further back and cite the example of Céline's Voyage au bout de la nuit? In your Fetishism of Archives, I thought the idea of the trace was important to be developed instead of relegated to a footnote; perhaps you could consider that. In lines 291/304 I thought you could expand your argument there. Furthermore, see my comment on lines 366-367 in a similar vein. 386-387 has awkward sentence phrasing. In your conclusion, I thought there were a couple of spots where your word choice seemed extreme, biased even, such as the "straitjacket of documents," etc. (See my comments.) Overall, it is very well written and presents a coherent argument that makes a fine contribution to scholarship. 

Author Response

dear anonymous reviewer,
thank you for your comments and suggestions for correction. I have followed some of them and tried to answer your doubts and those of the other reviewers. I enclose the corrected version of the article.

Reviewer 2 Report

The choice of topic of this paper is certainly timely: while the author(s) focus(es) on the French tradition of historiography in particular, the paper has vast implications for the discussion (and critique) of overlapping text types and reading and writing strategies.

I was slightly puzzled about the connection to the linguistic turn - or rather, it felt to me as if the linguistic turn might be misrepresented in this paper. To my understanding, the term "linguistic turn" denotes a series of different developments in Western thought in the 20th century, all of which are based on a fundamental skepticism about the idea that language is a transparent medium for grasping or conveying reality. Instead, this view is replaced by the view that language is an inescapable condition of thought. According to this view, all human cognition is structured by language; reality beyond language is regarded as "non-existent" in the sense that it may be an ontological reality, but is inaccessible. The reflection of thinking, especially philosophy, thus becomes a critique of language; a reflection of linguistic forms - also in literature - can thus only take place under the conditions of the reflected object - namely language. I fail to see, however, how turning research into literary forms and forgoing citations etc. can be an answer to this fundamental criticism - any chosen form will perpetuate it. For many a disciplineuding linguistics itself, increased scientific rigour, not narrative arbitrariness, has seemed the logical answer to it.

In that sense, I am not sure what the author(s) is/are critiquing here really is a manifestation of the linguistic turn. In many ways, it made me think more of the emotional turn which has been accused of supporting a partly anachronistic understanding of feelings with regard to emotions or emotional contexts evoked by historical documents.

When introducing the "literary" brand of historiography, the author(s) refer(s) to the described work as "experimental" (also in the abstract). I wasn't sure what the term "experiment" can add here, as it only evokes yet another, very "non-literary" bouquet of methods, or indeed the tradition of the literary experiment, which is, however, defined quite differently.

The author(s) also evoke(s) Coleridge's notion of "willing suspension of disbelief for the moment that constitutes poetic faith" as a sign of reading literature. Yet I got the impression that here, the suspension of disbelief may be more lasting - people may read (partly) fiction but actually believe to have learned about history. More importantly, not all literature comes with the expectation of fiction, which leads me to a more general criticism: for a literature journal, the terms literature, narrative and fiction are neither sufficiently differentiated, nor could I always pinpoint what the author(s) meant when they used them.

In terms of structure, I must admit to having found the introduction unclear and difficult to read; maybe the paper's hypotheses and assumptions could be made clearer. I also did not see the conclusion really integrating the findings of the paper to bring the argument to a close.

I would also suggest moving the section of the analytic part (where the author(s) locate(s) the representatives of the new historiography they describe) to the beginning of the section, so readers have a clearer idea of the breadth and implications of the trend.

I feel that the paper was a bit undecisive about discussing the pitfalls that come with abandoing scholarly rigour in favor of free narrative - the risk of anachronistic conclusions, misinterpretations etc. as called specifically by the representatives of the linguistic turn. I felt that a clearer stand regarding both potentials and risks of the approach would have been warranted.

Most of all, I will admit to having doubts about whether this paper is a good fit for Literature. While it focuses on the "literary" aspects of a particular brand of historiographic writing, it strikes me, in the end, as more of a history of science perspective on historiography than a contribution to literary scholarship.

Author Response

dear anonymous reviewer,
thank you for your comments and suggestions for correction. I have followed some of them and tried to answer your doubts and those of the other reviewers. I enclose the corrected version of the article.

I have clarified the "definition" of linguistic turn that I have chosen for my comments, underlining also the diversity of the interpretations proposed according to the authors and the disciplines. Contrary to what happens in linguistics and in other disciplines, the turn has weakened, in my opinion, the scientific bases of history. I specify this in a footnote.
As for the emotional turn: according to my knowledge, it means rather the entrance of emotions as a subject of history. Here, I mean the use of emotions of authors (historians) and readers as tools of validation of historical proof, or of restitution of the meaning of historical theses. I have tried to clarify this in my revision of the text.
I have also tried to clarify the vocabulary to better make explicit what I put behind the generic label "literature", fiction etc.: I hope this will be satisfactory for you.
I have modified a long paragraph at the end of the introduction to make my point clearer (I hope) and to distinguish between what comes back to the linguistic turn and what is new in these contemporary historiographic currents. And I have made this clearer in the conclusion as well.
As for your doubts about the relevance of the article in the journal, I understand, but I don't think it's up to me to answer them: it's rather the choice of the journal's curators. So I'm throwing the ball back to them!
A very big thank you for your work, and I hope to have satisfied your requirements.

Reviewer 3 Report

I recommend that this article be accepted in its current form.

Author Response

dear anonymous reviewer,
thank you for your work.
I have nevertheless reworked some passages of the article to answer the doubts of the other reviewers. I hope you will find it satisfactory too.

Author Response

Dear reviewer,
thank you for your work and your comments.
I have tried to respond to them, as well as to the remarks of others, by specifying the elements of the linguistic turn that seem relevant here, in my analysis of these current historiographic trends, without however going too deeply into the question: this is not really the object of my article.
I have also tried to make the last passages of the introduction clearer. And I have completed and revised the notes.
I hope you will agree with these changes.

Round 2

Reviewer 2 Report

Thank you for taking the time to revise the paper and for addressing some of my concerns - overall, I think the paper is quite improved.

I still have some concerns about the way you deal with the linguistic turn, which I'll try to make as plain as possible.

Let me start by saying that this is not a criticism of your argument, but rather one regarding presentation. I agree with you entirely: the linguistic turn has taken on very different forms in different disciplines, and its effects and outcomes also vary greatly. That being said, I still believe your paper would benefit (beyond a mere footnote) from a clearer distinction between definition, intentions and effects. I say this with the outlet in mind. Literature is primarily addressed at an audience from the fields of literary studies and its neighbouring disciplines. Hence, your readers will have in mind a definition of the linguistic turn that stems directly from analytic philosophy (hence my earlier point about 'scientific rigour'). It would help their understanding of your points immensely if you could include a paragraph where you first define the linguistic turn as historiography understands it; outline where it came from and why it evolved in the discipline; and then point out its actual effects and outcomes, combined with reflections on why they ended up, as you rightly say, nearly diametrically opposed to the intended scientific rigour that sparked the emergence of the turn in the first place.

Some of this information  is already in the paper, but it is distributed here and there. As I said, it would help greatly, if it were succinctly combined to make your point clearer. As it stands, it is quite difficult for a reader from a language/literature background to understand a) how the movement you describe even relates to the linguistic turn as they know it, b) how the original concept became the one that you describe, and c) how this particular understanding of the linguistic turn became a thing in historiography despite the fact that it apparently severely muddies the epistemological waters.

Author Response

Dear anonymous reviewer, thank you once again for your suggestions. I have tried to follow your advice. I have modified one paragraph by adding elements that I hope will clarify my position and the reasons why I introduced this discussion from Loriga and Revel's book into my analysis of current trends in French historiography. I don't think I can go any further without taking everything apart and going back otherwise. The linguistic turn is not the main point for me. I hope you will agree to keep the article as it is. 

In the attached version, you will find the changes highlighted in yellow. 

Best

Round 3

Reviewer 2 Report

Thank you for considering my points and adding some more revisions. I am now wholly satisfied and happy to recommend this paper for publication.